# A Mediterranean Diet Is Positively Associated with Bone and Muscle Health in a Non-Mediterranean Region in 25,450 Men and Women from EPIC-Norfolk

**DOI:** 10.3390/nu12041154

**Published:** 2020-04-21

**Authors:** Amy Jennings, Angela A. Mulligan, Kay-Tee Khaw, Robert N. Luben, Ailsa A. Welch

**Affiliations:** 1Norwich Medical School, Faculty of Medicine and Health Sciences, University of East Anglia, Norwich NR4 7UQ, UK; a.welch@uea.ac.uk; 2Strangeways Research Laboratory, Department of Public Health and Primary Care, Institute of Public Health, University of Cambridge, Worts Causeway, Cambridge CB1 8RN, UK; Angela.Mulligan@mrc-epid.cam.ac.uk (A.A.M.); kk101@medschl.cam.ac.uk (K.-T.K.); robert.luben@phpc.cam.ac.uk (R.N.L.); 3NIHR BRC Diet, Anthropometry and Physical Activity Group, MRC Epidemiology Unit, University of Cambridge, Cambridge CB2 0SL, UK

**Keywords:** Mediterranean diet, fracture, muscle

## Abstract

Research on Mediterranean diet (MD) adherence and musculoskeletal health is limited. The current study determined if adherence to the alternative MD score (aMED) and MD score (MDS), quantified from 7-d food diaries, was associated with fracture incidence, bone density (calcaneal broadband ultrasound attenuation (BUA)) and fat free mass (expressed over BMI (FFM^BMI^) using bioelectrical impedance) in 25,450 men and women recruited to the European Prospective Investigation into Cancer study in Norfolk, UK. During 17.4 years of follow up (443,178 total person years) 2195 incident fractures occurred. Higher aMED adherence was associated with 23% reduced total (Q5–Q1 HR 0.77; 95% CI 0.67, 0.88; *p*-trend < 0.01) and 21% reduced hip (Q5–Q1 HR 0.79; 95% CI 0.65, 0.96; *p*-trend = 0.01) fracture incidence, and significantly higher BUA (Q5–Q1 1.0 dB/MHz 95% CI 0.2, 1.9; *p*-trend < 0.01) and FFM^BMI^ (Q5–Q1 0.05 kg/(kg/m^2^) 95% CI 0.04, 0.06; *p*-trend < 0.01), comparing extreme adherence quintiles. Higher MDS was also associated with reduced total fractures (Q5–Q1 HR 0.83; 95% CI 0.71, 0.96; *p*-trend = 0.03) and significantly higher BUA (Q5–Q1 1.4 dB/MHz 95% CI 0.5, 2.3; *p*-trend < 0.01) and FFM^BMI^ (Q5–Q1 0.03 kg/(kg/m^2^) 95% CI 0.01, 0.04; *p*-trend < 0.01). This evidence supports the need to develop interventions to enhance MD adherence, particularly in women, where evidence for associations was stronger.

## 1. Introduction

A Mediterranean diet is widely recommended for the primary prevention of nutrition-related non communicable diseases including cardiovascular diseases, cancer and type 2 diabetes [1,2]. Data from randomised controlled trials and prospective cohort studies also show that a Mediterranean diet may be protective for musculoskeletal health [3,4] although the research is limited [5]. A Mediterranean diet is characterised by high intake of plant-based foods (fruit, vegetables, nuts and cereals) and olive oil, moderate intake of fish and alcohol and low intake of dairy products and red and processed meat [6]. Nutritional constituents of foods abundant in Mediterranean diets such as antioxidants (including vitamin C, carotenoids, selenium, magnesium) and fibre (abundant in plant foods) have been associated with beneficial effects on musculoskeletal health [7]. Likewise, polyphenols (abundant in fruit, vegetables and olive oil), and omega-3 fatty acids (abundant in fish) have been shown to protect against bone loss [8,9]. The pathways of action for antioxidants and polyphenols in bone remodelling include suppressing osteoclast activity and promoting osteoblast differentiation [10]. The n-3 and n-6 omega-3 fatty acids have been shown to inhibit inflammatory cytokine expression, stimulate prostaglandin E2 production and enhance calcium transport [11]. Whilst individual dietary constituents within the Mediterranean diet have an important role in musculoskeletal health, it is likely the interactive and synergistic effects of these individual components have a greater combined impact [12].

It is well established that both low bone mass and skeletal muscle mass increase susceptibility to low impact fragility fractures. Furthermore, skeletal muscle and bone health appear to be related with bone tissue shown to be responsive to alterations in skeletal muscle contraction and myokines secreted by the muscle, including insulin-like growth factor 1 and fibroblast growth factor 2, stimulating bone growth and repair [13,14]. This means that a decline in muscle mass is likely to have a corresponding impact on bone health. The aim of the current study was therefore to examine associations between adherence to a Mediterranean diet and key indicators of bone and muscle status, including fracture incidence, bone density and fat free mass in a large longitudinal cohort of 25,450 men and women living in the UK. We also examined the associations between our bone and muscle status indicators and fracture risk in order to ascertain their value for prediction of fracture risk with a longer follow-up period than previously reported. Earlier studies examining the associations between Mediterranean diet adherence, bone density and fat free mass (FFM) have been restricted by limited sample sizes [15,16,17]. In addition, this will be the first study to use food diaries to examine Mediterranean diet adherence in a large longitudinal study of fracture risk.

## 2. Materials and Methods

### 2.1. Study Population

The European Prospective Investigation into Cancer (EPIC) in Norfolk is a population-based prospective cohort of 25,450 men and women in the UK arm of the multicentre EPIC study (Figure 1). The EPIC Norfolk study design and the characteristics of the cohort have been described previously [18]. Briefly, all men and women aged between 39 to 79 years between 1993 and 1997 who were registered with 35 collaborating general practice surgeries in the Norfolk area were invited to the study by mail and attended a baseline health check. The study was approved by the Norwich District Authority Ethics Committee (98CN01) and was conducted according to the Declaration of Helsinki. All participants provided informed consent.

### 2.2. Outcome Assessment

Of the recruited participants, 14,815 agreed to take part in a follow-up examination between 1998 and 2000 (second health check), at which point quantitative ultrasound measurements of the calcaneus was performed using a CUBA Clinical Ultrasonometer (McCue Ultrasonics). Measurements of broadband ultrasound attenuation (BUA in dB/MHz) and velocity of sound (VOS in m/s) were taken in duplicate from each foot. The means of the left and right measurements were used for analysis. The coefficient of variation was 3.5%. Five CUBA machines were used and calibrated daily with a physical phantom. The machines were also compared on one calcaneus, and whilst significant variations between machines arose, adjustment for temperature and machine were not found to change BUA–fracture relations [19]. Analysis in this cohort also showed that the power of quantitative ultrasound for the prediction of fractures is comparable to that of dual-energy x-ray absorptiometry (DXA) with the hazard ratio for a 1 SD decrease in DXA-measured total hip bone mineral density was 2.3 (95% CI: 1.7–3.0) compared with 2.0 (95% CI: 1.6–2.7) for a 1 SD decrease in BUA [20].

Bioelectrical impedance analysis (BIA) was also undertaken using a standardised technique (Tanita TBF-531, Bodystat, Isle of Man, UK). The machine had two toe and two heel electrodes. Body density (BD) was calculated from weight, height, and impedance (Z) using standard regression formulas for adults:
BD in men = 1.100455 − 0.109766 × weight (kg) × Z/height (cm)^2^ + 0.000174 × Z
BD in women = 1.090343 − 0.108941 × weight (kg) × Z/height (cm)^2^ + 0.00013 × Z.


From this, fat free mass (FFM) in kg was calculated:
FFM = weight (kg) − ((4.57/BD − 4.142) × weight (kg))


This estimates the total mass of non-fat compartments of the body, i.e., metabolic tissue, intra- and extra-cellular water, and bone tissue. For analysis, we standardised FFM by dividing values by BMI (FFM^BMI^) [21]. The correlation between DXA and BIA-calculated fat free mass is high (0.95) [22].

Osteoporotic fractures were identified using the East Norfolk Health Authority database which records all hospital contact that Norfolk residents have in England and Wales. International Classification of Disease codes 9 and 10 diagnostic codes were used to ascertain fractures by site. Data were available to 31st March 2016. We defined total fractures as any incident fracture at the hip, spine or wrist.

### 2.3. Assessment of Dietary Intakes

Food and nutrient intakes were estimated from seven-day food diaries completed at one time point (first health check (1993–1997), as described previously [23]. For the first day of the diary, a nurse conducted a 24-h diet recall according to a standardised protocol, with the remaining six days completed by the participant at home. Data were entered by using the Data into Nutrients for Epidemiologic Research (DINER) data entry system [24]. We calculated adherence to the original Mediterranean Diet Score (MDS) developed by Trichopoulou et al. [6] and the alternate Mediterranean Diet Index (aMED), as developed by Fung et al. [25]. The indices are based on the dietary intake of nine components (MDS: vegetables, legumes, fruit and nuts, dairy, cereals, meat and meat products, fish, alcohol, and the ratio of monounsaturated to saturated fat and aMED: vegetables, legumes, fruit, nuts, whole grains, red and processed meat, fish, alcohol, and the ratio of monounsaturated fat to saturated fat). Each component receives one point if intake is above the sex-specific median except for meat and, for the MDS only, dairy (where one point is scored if consumption is less than the median intake). For alcohol, one point is given for intake between 5–25 g/d in the aMED and MDS (women only) or between 10–50 g/day for men in the MDS. The final scores range between 0 and 9 with a higher score implying greater adherence. The key difference between these two indices of Mediterranean diet adherence is the scoring of dairy products, which are scored negatively in the MDS and eliminated in the aMED. As a result, we expected that the nutrient profiles of high MDS and aMED adherers would vary—in particular, calcium and protein intake, which are rich in dairy products.

### 2.4. Assessment of Covariates

Participants completed a Health and Lifestyle Questionnaire at each health check which included questions on past year physical activity, from which a simple four-level physical activity index was derived [26], smoking status, medication use, menopausal status and family history of medical conditions. Trained nurses measured height and weight, from which BMI was calculated. We estimated the ratio of energy intake to estimated energy requirements (EER) (100 × (energy intake (kcal)/EER) with EER determined using the Institute of Medicine equations by BMI, age and sex [27].

### 2.5. Statistical Analysis

The characteristics of participants were presented as means (SD) and *n* (%) for categorical variables and compared between men and women using independent t-tests or chi-square tests at the first (1993–1997) and second (1998–2000) health checks. We examined the associations between aMED and MDS adherence and dietary intakes at the first health check using ANCOVA with sex, age, BMI, smoking, physical activity, days of dietary intake data, and the ratio of energy intake to EER (as a measure of energy misreporting) included in the models. Multivariable cox proportional-hazard models were used to determine the associations between incident fractures and measures of quantitative ultrasound (BUA and VOS) and FFM^BMI^ with sex, age, BMI, smoking, physical activity, family history of osteoporosis, medication use (corticosteroids, aspirin or hormone replacement therapy) and menopausal status included as covariates. We then modelled the associations between aMED and MDS adherence and incident fractures with additional adjustment for calcium intake, supplement use (vitamin D or calcium), days of dietary intake data and the ratio of energy intake to EER as a measure of energy misreporting, both in the whole cohort and stratified by sex. Models with the same adjustments were used to examine associations between individual dietary components of the aMED and MDS scores and incident fracture. In these analyses, we compared participants above and below median intakes and included all components. Finally, we investigated the associations between aMED and MDS adherence and BUA, VOS and FFM^BMI^ at the second health check using ANCOVA with adjustment for all covariates (sex, age, BMI, smoking, physical activity, family history of osteoporosis, medication use, menopausal status, calcium intake, supplement use (vitamin D or calcium), days of dietary intake data and the ratio of energy intake to EER). Participants were ranked into quintiles of aMED and MDS adherence for all analyses. Tests for trends were conducted using the median value for each quintile as a continuous variable. *p* values < 0.05 was considered statistically significant. Statistical analyses were performed using Stata software (version 16; College Station, TX, USA: StataCorp LLC).

## 3. Results

Our analysis included 25,453 participants aged between 39 and 79 y at baseline (mean 59.3 ± 9.3 years), of which 55% were women (Table 1). During follow up (mean 17.4 y; 443,178 total person years) there were 2195 incident spine, wrist and hip fractures (*n* = 467 men and *n* = 1167 women), of which 49% were hip fractures. Women had a 6.1% higher total fracture incidence than men (*p* < 0.01). Measurements of BUA, VOS and FFM^BMI^ were taken at the second health check when participants were 62.2 years old (± 9.0, range 42–82) and all were significantly lower in women compared to men (*p* < 0.01 for all).

There were significant trends for lower fracture risk across increasing quintiles of BUA (Q5 vs. Q1 HR 0.32; 95% CI 0.26, 0.40; *p*-trend < 0.01), VOS (Q5 vs. Q1 HR 0.33; 95% CI 0.26, 0.41; *p*-trend < 0.01) and FFM^BMI^ (Q5 vs. Q1 HR 0.56 95% CI 0.43, 0.74; *p*-trend < 0.01) after adjustment for sex, age, BMI, smoking, physical activity, family history of osteoporosis, medication use and menopausal status (Figure 2).

Higher aMED and MDS adherence was associated with significantly higher intakes of energy (aMED Q5 vs. Q1 13.0 kcal; 95% CI 7.3, 18.7; *p*-trend < 0.01, MDS Q5 vs. Q1 17.7 kcal; 95% CI 11.5, 23.9; *p*-trend < 0.01), vitamin C (aMED Q5 vs. Q1 42.1 mg; 95% CI 40.2, 44.1; *p*-trend < 0.01, MDS Q5 vs. Q1 33.8 mg; 95% CI 31.6, 35.9; *p*-trend < 0.01), magnesium (aMED Q5 vs. Q1 78.6 mg; 95% CI 76.1, 81.1; *p*-trend < 0.01, MDS Q5 vs. Q1 46.3 mg; 95% CI 43.5, 49.2; *p*-trend < 0.01) and potassium (aMED Q5 vs. Q1 656 mg; 95% CI 633, 678; *p*-trend < 0.01, MDS Q5 vs. Q1 380 mg; 95% CI 355, 406; *p*-trend < 0.01) (Appendix A). Energy adjusted intakes of saturated (aMED Q5 vs. Q1 −3.3%; 95% CI −3.4, −3.2; *p*-trend < 0.01, MDS Q5 vs. Q1 −3.5%; 95% CI −3.6, −3.4; *p*-trend < 0.01) and monounsaturated fats (aMED Q5 vs Q1 −0.6%; 95% CI −0.6, −0.5; *p*-trend < 0.01, MDS Q5 vs. Q1 −0.7%; 95% CI −0.8, −0.6; *p*-trend < 0.01) were lower with increasing adherence, although the ratios of monounsaturated to saturated fats and polyunsaturated fat intakes were higher (aMED Q5 vs. Q1 0.20; 95% CI 0.19, 0.21; *p*-trend < 0.01, MDS Q5 vs. Q1 0.20; 95% CI 0.19, 0.21; *p*-trend < 0.01). Higher adherence to the MDS, which scores dairy products negatively, was associated with significantly lower calcium intakes (Q5 vs. Q1 −78.9 mg; 95% CI −88.3, −69.5; *p*-trend < 0.01) with no difference observed for protein intake (Q5 vs. Q1 −0.1 %; 95% CI −0.2, 0.0; *p*-trend = 0.19). Conversely, higher aMED adherence, which does not account for dairy intake, was associated with higher intakes of calcium (Q5 vs. Q1 23.3 mg; 95% CI 14.5, 32.0; *p*-trend < 0.01) and protein (Q5 vs. Q1 0.9%; 95% CI 0.8, 1.0; *p*-trend < 0.01).

In multivariable analysis, in the whole cohort there was a significant inverse trend for reduced total fracture risk across increasing quartiles of aMED adherence (Q5 vs. Q1 HR 0.77; 95% CI 0.67, 0.88; *p*-trend < 0.01) (Figure 3) adjusted for relevant confounding variables.

After stratification by sex, these associations were only apparent in women (Q5 vs. Q1 HR 0.75; 95% CI 0.61, 0.91; *p*-trend < 0.01) (Figure 4). In analysis restricted to participants aged over 50 years, who experienced 92% of fractures, we observed a significant inverse trend for reduced fracture risk by higher aMED score (Q5 vs. Q1 HR 0.83; 95% CI 0.72, 0.95; *p*-trend < 0.01) (Figure 3). Likewise, we observed a significant trend for reduced fracture risk by higher aMED score in post-menopausal women (Q5 vs. Q1 HR 0.85; 95% CI 0.71, 1.02; *p*-trend = 0.04) (Appendix A). The unadjusted hazard ratios for all participants by Mediterranean Diet adherence are presented in Appendix A. Higher adherence to the MDS was also associated with reduced fracture risk in the whole cohort (Q5 vs. Q1 HR 0.83; 95% CI 0.71, 0.96; *p*-trend = 0.03), there were no associations with MDS when the cohort was stratified by sex or age. For hip fracture, we found a significant inverse trend for aMED (Q5 vs. Q1 HR 0.79; 95% CI 0.65, 0.96; *p*-trend = 0.01) but not MDS (Q5 vs. Q1 HR 0.85; 95% CI 0.69, 1.04; *p*-trend = 0.24) adherence (data not shown).

Of the individual components of the aMED score, intakes above the median of legumes (HR 0.91; 95% CI 0.84, 0.99; *p*-trend = 0.04), nuts and seeds (HR 0.86; 95% CI 0.77, 0.96; *p*-trend = 0.01), and alcohol intake (HR 0.91; 95% CI 0.83, 0.99; *p*-trend = 0.04) were associated with reduced fracture risk in mutually adjusted models, whereas intake below the median of red and processed meat (HR 0.91; 95% CI 0.83, 0.99; P-trend = 0.03) was associated with reduced fracture risk. No associations were observed for individual components of the MDS (Table 2).

We also observed significant positive trends between higher BUA and VOS across increasing quintiles of aMED adherence (Q5 vs. Q1 BUA = 1.0 dB/MHz 95% CI 0.2, 1.9; *p*-trend < 0.01 and VOS = 2.3 m/s 95% CI 0.3, 4.2; *p*-trend < 0.01, Table 3). Stratification by sex revealed significant associations between aMED adherence and BUA only in women (Q5 vs. Q1 1.2 dB/MHz 95% CI 0.2, 2.2; *p*-trend < 0.01). Likewise, there was a significant positive association between higher aMED adherence and FFM^BMI^ (Q5 vs. Q1 0.05 kg/(kg/m^2^) 95% CI 0.04, 0.06; *p*-trend < 0.01) which was observed in both men (Q5 vs Q1 0.05 kg/(kg/m^2^) 95% CI 0.03, 0.06; *p*-trend < 0.01) and women (Q5 vs. Q1 0.05 kg/(kg/m^2^) 95% CI 0.03, 0.07; *p*-trend < 0.01). The associations between aMED adherence and FFM expressed in kg were unchanged ((Q5 vs. Q1 0.92 kg 95% CI 0.65, 1.19; *p*-trend < 0.01, Appendix A). The unadjusted associations between BUA, VOS and FFM^BMI^ and aMED adherence are presented in Appendix A.

For the MDS, there was a significant positive trend between higher adherence and higher BUA in the whole cohort (Q5 vs. Q1 1.4 dB/MHz 95% CI 0.5, 2.3; *p*-trend < 0.01) and in women (Q5 vs. Q1 0.9 dB/MHz 95% CI −0.2, 1.9; *p*-trend = 0.02) (Table 4). There was also a significant positive association between higher MDS adherence and FFM^BMI^ in the whole cohort (Q5 vs. Q1 0.03 kg/(kg/m^2^) 95% CI 0.01, 0.04; *p*-trend < 0.01) and in both men (Q5 vs. Q1 0.02 kg/(kg/m^2^) 95% CI 0.01, 0.05; *p*-trend = 0.02) and women (Q5 vs. Q1 0.03 kg/(kg/m^2^) 95% CI 0.01, 0.05; *p*-trend < 0.01). There were also significant associations between higher MDS adherence and FFM expressed in kg ((Q5 vs. Q1 1.21 kg 95% CI 0.92, 1.51; *p*-trend < 0.01, Appendix A). We observed no significant associations between MDS adherence and VOS. The unadjusted associations between BUA, VOS and FFM^BMI^ and MDS adherence are presented in Appendix A.

## 4. Discussion

In this large sample of men and women living in a non-Mediterranean region, greater adherence to a Mediterranean diet was associated with up to 23% reduced total and 21% reduced hip fracture incidence comparing extreme quintiles, with a one point increase in aMED and MDS adherence corresponding to a 4% and 3% reduction in total fracture incidence, respectively. These associations were greater in women than men and independent of age, BMI, smoking, physical activity, menopausal status and medication use including hormone replacement therapy. In addition, we found that greater Mediterranean diet adherence was associated with significantly higher BUA (1.4 dB/MHz), VOS (2.3 m/s) and FFM^BMI^ (0.05 kg/(kg/m^2^)). To our knowledge this was the first study to use food diaries to examine Mediterranean diet adherence in a large longitudinal study of fracture risk [4,5,15,28,29]. Additional findings that higher BUA and VOS were associated with reduced fracture risk have previously been reported in this cohort over a shorter follow-up period however, the finding that higher FFM predicts lower fracture risk is novel [19].

These findings are consistent with those of the Women’s Health Initiative study in the US where postmenopausal women scoring in the highest quintile of aMED adherence had 20% lower risk of hip fracture, although no association between aMED and total fractures was observed [15]. Likewise, the Consortium on Health and Ageing: Network of Cohorts in Europe and the United States reported a two-point increment in adherence to the MDS corresponded to a 4% reduction in hip fracture incidence [4]. Analysis of EPIC-Europe cohorts showed the highest quartile of MDS adherence to be associated with a 27% reduction in hip fracture risk, equivalent to a 7% reduction per one-point increase in MDS [28]. In the Nurses’ Health Study there was a 14% lower risk of hip fractures in the highest compared to the lowest quintile of aMED adherence [29]. As the effect sizes in the current study are in line with these previous reports using food frequency questionnaires to measure Mediterranean diet adherence [4,15,28,29], it appears that dietary assessment method is not a significant factor in determining these associations. This indicates, therefore, that the overall distribution of intakes is more important than absolute intake values when classifying Mediterranean diet adherence within a population.

Our previous research in a random subset of this cohort (*n* = 5319) with a mean follow-up time <13.4 years showed no significant trends for total fracture risk by quintile of vitamin C, magnesium or potassium intake, although alpha- and beta-carotene intakes were associated with up to 29% lower risk of fractures in men only [30,31,32]. Whilst these dietary constituents are important components of the Mediterranean diet and have mechanistic links with musculoskeletal health, this research suggests that the synergistic effects of these, and other individual components, are likely to be of greater importance in reducing fracture risk. Previous research has shown that the food matrix exhibits distinct relations with health outcomes compared to single nutrients; for example, whole dairy products have been shown to have a greater effect on bone mass than single dairy constituents, such as calcium and vitamin D [33].

A particular strength of the current study was the inclusion of data on musculoskeletal risk factors, including bone density and FFM, for 14,815 participants. Comparing extreme quintiles of these risk factors revealed a 68%, 67% and 44% reduction in total fractures for BUA, VOS and FFM, respectively (mean follow-up time 17.4 years). Likewise, extreme quintiles of BUA and VOS were associated with a 69% and 67% reduction in hip fractures, with no significant reductions by FFM. Earlier research in this cohort with a mean follow-up of 1.9 years reported that a 1 SD difference in BUA (equivalent to 20 db/MHz) and VOS (equivalent to 40 m/s) was associated with a relative risk of fracture of 1.95 and 1.63, respectively [19] and analysis in a subset (*n* = 1151) over 10 years showed that a 1 SD decrease in BUA was associated with a hazard ratio of 2.0 [20]. Together, research in this cohort suggests that maintaining BUA, VOS and FFM is an important preventative target for fracture risk.

Previous research on these particular risk factors (BUA, VOS and FFM) and Mediterranean diet adherence is limited. A study of 418 Italian healthy men and women showed positive associations between higher Mediterranean diet adherence and higher quantitative ultrasound index [34]. In 198 university students, adherence to the Mediterranean Diet Quality Index was associated with higher FFM assessed using BIA [35]. Analysis of 7961 women recruited to the Women’s Health Initiative study found no associations between aMED adherence and bone mineral density or lean body mass measured using dual-energy x-ray absorptiometry (DXA) [15], although our previous analysis of 2570 women from the TwinUK cohort showed higher MDS adherence was associated with lower fat free mass measured using DXA [16] and analysis of 2371 Chinese adults found higher aMED scores were positively and dose-dependently associated with DXA-measured bone mineral density at multiple sites [17].

Although adherence to both the aMED and MDS was associated with reduced fracture risk, the effect sizes were stronger for aMED adherence. One of the key differences between the scores was differential scoring of dairy intakes, which resulted in higher calcium intakes with high adherence to the aMED and lower calcium intakes with higher MDS adherence. However, notably, when we examined the individual components of the score, dairy intake did not appear to be a significant factor in predicting fracture risk in this cohort. We suggest that future studies detail the components of the MDS used to ensure the appropriate interpretation of their results.

Strengths of our study include the wide age range of both men and women followed for 17 years and the large number of fractures (*n* = 2195) ascertained without self-report. This was the first study to use food diaries to measure Mediterranean diet adherence and although our results are comparable to fracture studies using food frequency questionnaires this provides useful information for future studies planning to measure diet quality scores. In addition, this was, to our knowledge, the largest cross-sectional study to examine associations between Mediterranean diet adherence, bone density and FFM in both men and women. Whilst DXA measured is regarded at the gold standard assessment method for bone density and body composition, recent research suggests that whilst BIA overestimates FFM compared to DXA, the methods are interchangeable at a population level [36] and calcaneus ultrasonography is predictive of bone quality as measured by DXA [37] and produced comparable fracture risk predictions as DXA in this cohort [20]. In addition, we were able to relate FFM to fracture risk for the first time in this cohort therefore demonstrating the potential importance of FFM as a preventative target for fracture risk. Despite the strengths of this study, there are also some limitations, the food diaries were completed once at baseline and we cannot exclude the possibility that dietary changes occurred during follow-up. Although we expected that the number of high-trauma fractures would be low, we were not able to exclude these cases. We also had no functional measures of sarcopenia which, alongside muscle mass, is an important element of disease definition [38]. Finally, even though we adjusted for a range of important confounding factors, we cannot exclude the possibility of residual confounding.

In conclusion, we have shown for the first time that Mediterranean diet adherence assessed using food diaries is associated with fracture risk at a similar magnitude to studies using food frequency questionnaires. We found that few of the individual components of the Mediterranean diet adherence scores were associated with fracture risk, indicating that in this non-Mediterranean region the additive effects of individual components within the Mediterranean diet have a greater effect. This evidence supports the need to develop interventions to enhance Mediterranean diet adherence, particularly in women, where evidence for associations was stronger.

## Figures and Tables

**Figure 1 nutrients-12-01154-f001:**
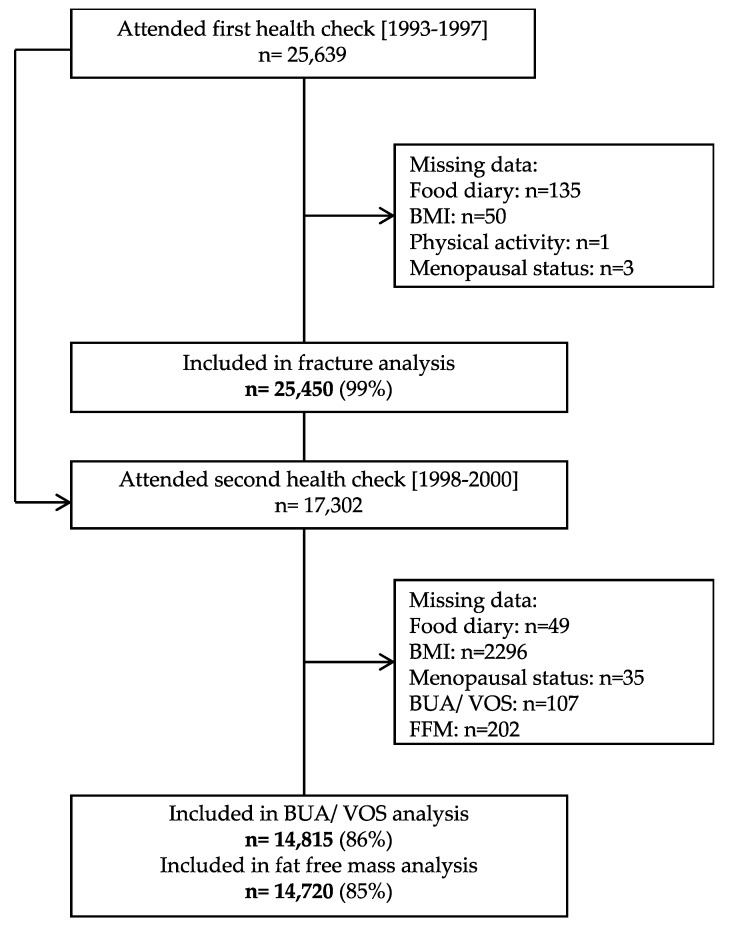
A flow chart detailing participant recruitment for the European Prospective Investigation of Cancer.

**Figure 2 nutrients-12-01154-f002:**
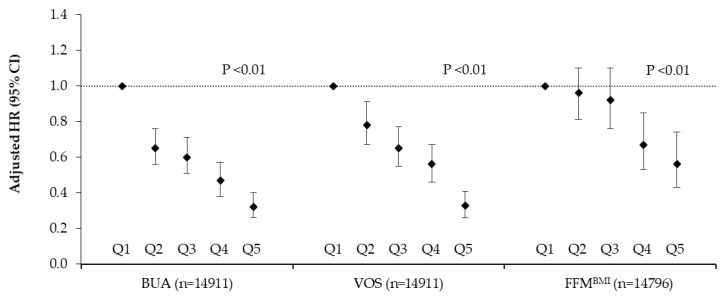
Total fracture risk after a mean follow up of 17.6 years by quintile of broadband ultrasound attenuation, velocity of sound and fat free mass in 14,911 men and women aged 39–79 years. Values are adjusted hazard ratios (95% CI), *n* = 14,911. Ratios were adjusted for sex, age, BMI, smoking, physical activity, family history of osteoporosis, medication use (corticosteroids, aspirin or hormone replacement therapy) and menopausal status. BMI was not included in the FFM^BMI^ model. *p* values are for trends calculated using the Cox proportional hazards model. Participant numbers (cases) per quintile were as follows; BUA: Q1 = 2985 (505), Q2 = 2981 (266), Q3 = 2981 (213), Q4 = 2982 (157), Q5 = 2982 (105). VOS: Q1 = 2994 (464), Q2 = 2989 (292), Q3 = 2991 (224), Q4 = 2972 (172), Q5 = 2965 (94). FFM^BMI^: Q1 = 2960 (308), Q2 = 2959 (325), Q3 = 2959 (270), Q4 = 2959 (182), Q5 = 2959 (145). BUA = broadband ultrasound attenuation; VOS = velocity of sound; FFM^BMI^ = fat free mass adjusted for BMI

**Figure 3 nutrients-12-01154-f003:**
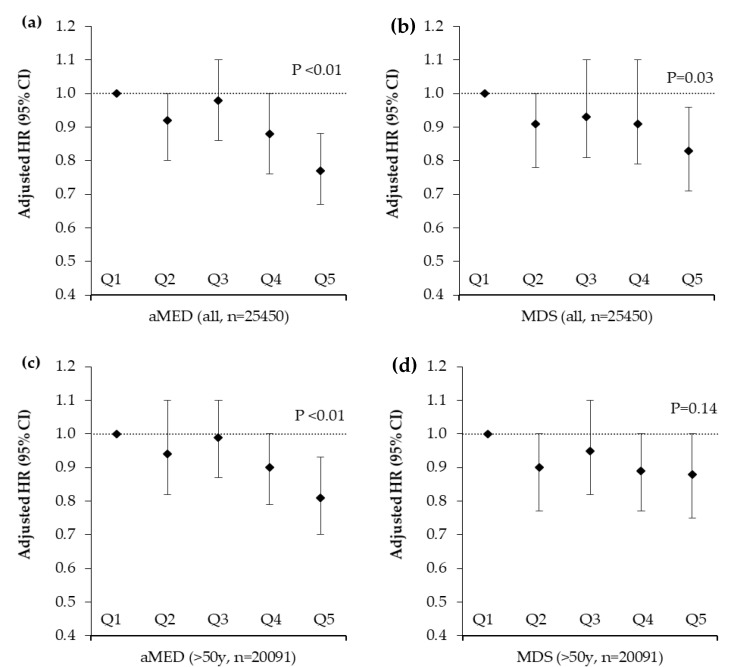
Total fracture risk after a mean follow up of 17.6 years by quintile of alternative Mediterranean Diet Score and Mediterranean Diet Score in 25,450 men and women aged 39–79 years stratified by age. Values are adjusted hazard ratios (95% CI), *n* = 25,450. Ratios were adjusted for sex, age, BMI, smoking, physical activity, family history of osteoporosis, calcium intakes, supplement use (vitamin D or calcium), medication use (corticosteroids, aspirin or hormone replacement therapy), menopausal status, days of dietary intake data and the ratio of energy intake to estimated energy requirements. *p* values are for trends calculated using the Cox proportional hazards model. Participant numbers (cases) per quintile were as follows; (**a**) aMED all participants: Q1 = 5161 (481), Q2 = 4991 (430), Q3 = 5557 (515), Q4 = 4740 (396), Q5 = 5001 (373) (**b**) MDS all participants: Q1 = 3304 (317), Q2 = 4779 (416), Q3 = 5918 (528), Q4 = 5681 (494), Q5 = 5768 (440) (**c**) aMED participants >50 years: Q1 = 4151 (446), Q2 = 3983 (401), Q3 = 4422 (470), Q4 = 3688 (364), Q5 = 3847 (339) (**d**) MDS participants >50 years: Q1 = 2641 (296), Q2 = 3848 (380), Q3 = 4668 (495), Q4 = 4478 (436), Q5 = 4456 (413). aMED = alternative Mediterranean Diet Score, MDS = Mediterranean Diet Score.

**Figure 4 nutrients-12-01154-f004:**
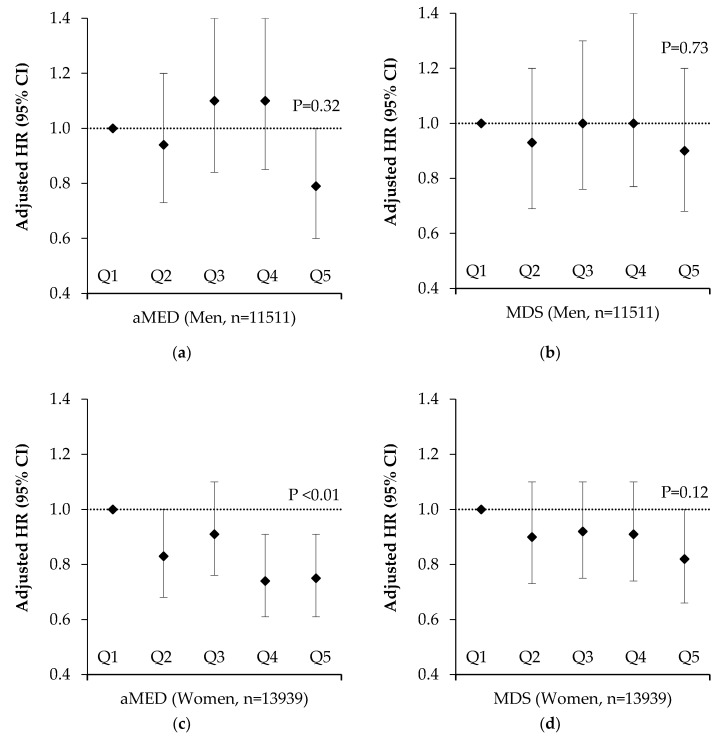
Total fracture risk after a mean follow up of 17.6 years by quintile of alternative Mediterranean Diet Score and Mediterranean Diet Score in 25,450 men and women aged 39–79 years stratified by sex. Values are adjusted hazard ratios (95% CI), *n* = 25,450. Ratios were adjusted for age, BMI, smoking, physical activity, family history of osteoporosis, calcium intakes, supplement use (vitamin D or calcium), medication use (corticosteroids, aspirin or hormone replacement therapy), menopausal status, days of dietary intake data and the ratio of energy intake to estimated energy requirements. *p* values are for trends calculated using the Cox proportional hazards model. Participant numbers (cases) per quintile were as follows; (**a**) aMED Males: Q1 = 2300 (121), Q2 = 2236 (111), Q3 = 2522 (145), Q4 = 2162 (131), Q5 = 2291 (102) (**b**) MDS Males: Q1 = 1397 (75), Q2 = 2070 (103), Q3 = 2637 (145), Q4 = 2626 (150), Q5 = 2781 (137) (**c**) aMED Females: Q1 = 2861 (360), Q2 = 2755 (319), Q3 = 3035 (370), Q4 = 2578 (265), Q5 = 2710 (271) (**d**) MDS Females: Q1 = 1907 (242), Q2 = 2709 (313), Q3 = 3281 (383), Q4 = 3055 (344), Q5 = 2987 (303). aMED = alternative Mediterranean Diet Score, MDS = Mediterranean Diet Score.

**Table 1 nutrients-12-01154-t001:** Characteristics of participants in the EPIC Norfolk study at the first (*n* = 25,450) and second (*n* = 14,815) health checks, stratified by sex ^1^.

	1st Health Check (1993–1997)				2nd Health Check (1998–2000)			
	All (*n* = 25,450)	Men (*n* = 11,511)	Women (*n* = 13,939)	*p*=	All (*n* = 14,815)	Men (*n* = 6490)	Women (*n* = 8325)	*p*=
Age, years	59.3 ± 9.3	59.7 ± 9.3	58.9 ± 9.3	<0.01	62.2 ± 9.0	62.9 ± 9.0	61.6 ± 9.0	<0.01
Fracture, y	2195 (8.6)	610 (5.3)	1585 (11.4)	<0.01	-	-	-	-
BUA, dB/MHz	-	-	-	-	79.9 ± 19.2	90.1 ± 17.5	72.1 ± 16.5	<0.01
VOS, m/s	-	-	-	-	1634 ± 41.4	1645 ± 39.9	1625 ± 40.2	<0.01
BMI, kg/m^2^	26.4 ± 3.9	26.5 ± 3.3	26.2 ± 4.4	<0.01	26.7 ± 4.0	26.9 ± 3.3	26.5 ± 4.4	<0.01
FFM (kg)	-	-	-	-	49.8 ± 11.8	61.7 ± 6.0	40.6 ± 4.8	<0.01
FFM^BMI^ (kg/(kg/m^2^))	-	-	-	-	1.9 ± 0.5	2.3 ± 0.3	1.6 ± 0.3	<0.01
Current smoking, y	3170 (12.5)	1472 (12.8)	1698 (12.2)	<0.01	1243 (8.5)	540 (8.4)	703 (8.6)	<0.01
Physically active, y	4610 (18.1)	2479 (21.5)	2131 (15.3)	<0.01	2825 (19.4)	1431 (22.4)	1394 (17.0)	<0.01
Family history of osteoporosis, y	1117 (4.4)	307 (2.7)	810 (5.8)	<0.01	686 (4.7)	174 (2.7)	512 (6.3)	<0.01
Vitamin D supplements, y	6849 (26.9)	2570 (22.3)	4279 (30.7)	<0.01	4335 (29.7)	1599 (25.0)	2736 (33.5)	<0.01
Calcium supplements, y	911 (3.6)	165 (1.4)	746 (5.4)	<0.01	598 (4.1)	98 (1.5)	500 (6.1)	<0.01
Corticosteroid or aspirin, y	2687 (10.6)	1508 (13.1)	1179 (8.5)	<0.01	1667 (11.4)	883 (13.8)	784 (9.6)	<0.01
HRT, y	-	-	2824 (20.3)	-	-	-	1732 (21.2)	-
Postmenopausal, y	-	-	8351 (59.9)	-	-	-	6002 (73.4)	-
aMED, points	4.0 ± 1.7	4.0 ± 1.7	4.0 ± 1.7	0.19	4.1 ± 1.7	4.1 ± 1.7	4.1 ± 1.7	0.02
MDS, points	4.3 ± 1.6	4.4 ± 1.6	4.2 ± 1.6	<0.01	4.33 ± 1.59	4.40 ± 1.60	4.28 ± 1.58	<0.01
Energy intake, kcal/d	1941 ± 534	2240 ± 527	1694 ± 395	<0.01	1974 ± 517	2285 ± 502	1731 ± 380	<0.01
Calcium intakes, mg/d	835 ± 282	919 ± 297	766 ± 248	<0.01	853 ± 275	942 ± 289	784 ± 243	<0.01
Vitamin D intakes, mg/d	3.3 ± 2.4	3.7 ± 2.7	2.9 ± 2.1	<0.01	3.4 ± 2.4	3.9 ± 2.8	3.0 ± 1.9	<0.01

**^1^** Values are means ± SD or *n* = (%), *n* = 25,438. *p* values are for the difference between men and women using *t*-test for continuous variables and χ^2^ for categorical variables. Fat free mass was measured in *n* = 14,720 (*n* = 6446 men and *n* = 8274 women) at the second health check. aMED = Alternative Mediterranean Diet Score; BUA = broadband ultrasound attenuation; FFM = fat free mass; HRT = hormone replacement therapy; MDS = Mediterranean Diet Score; VOS = velocity of sound; y=yes.; d = day.

**Table 2 nutrients-12-01154-t002:** Total fracture risk after a mean follow up of 17.6 years by individual components of the alternative Mediterranean Diet Score and Mediterranean Diet Score in 25,450 men and women aged 39–79 years ^1^.

Score	Component	Hazard Ratio (95% CI)	*p*-Value
**aMED**	Vegetables, g/d	0.93 (0.86, 1.02)	0.13
	Legumes, g/d	0.91 (0.84, 1.00)	0.04
	Fruit, g/d	0.97 (0.89, 1.06)	0.53
	Fish, g/d	1.01 (0.93, 1.11)	0.76
	Nuts and seeds, g/d	0.86 (0.77, 0.96)	0.01
	Wholegrains, g/d	1.01 (0.92, 1.10)	0.89
	Red and processed meat, g/d	0.91 (0.83, 0.99)	0.03
	Ratio MUFA: SFA	1.03 (0.94, 1.12)	0.57
	Alcohol, g/d	0.91 (0.83, 0.99)	0.04
**MDS**			
	Vegetables, g/d	0.94 (0.86, 1.02)	0.14
	Legumes, g/d	0.92 (0.84, 1.00)	0.06
	Fruit and nuts, g/d	0.97 (0.89, 1.06)	0.54
	Fish, g/d	1.01 (0.92, 1.10)	0.85
	Cereals, g/d	0.99 (0.90, 1.09)	0.82
	Dairy, g/d	0.94 (0.84, 1.05)	0.28
	Meat and eggs, g/d	0.96 (0.88, 1.05)	0.38
	Ratio MUFA: SFA	1.01 (0.93, 1.11)	0.76
	Alcohol, g/d	0.93 (0.85, 1.02)	0.13

**^1^** Values are the adjusted hazard ratios (95% CI) comparing participants with intakes above vs. below the median for each component (except the Meat and Dairy components, which compare participants with intakes below vs. above the median). Ratios were adjusted for sex, age, BMI, smoking, physical activity, family history of osteoporosis, calcium intakes, supplement use (vitamin D or calcium), medication use (corticosteroids, aspirin or hormone replacement therapy), menopausal status, days of dietary intake data and the ratio of energy intake to estimated energy requirements. As each component was included simultaneously in the same model, values are mutually adjusted for the other components. d: day.

**Table 3 nutrients-12-01154-t003:** Measures of broadband ultrasound attenuation, velocity of sound and fat free mass by quintile of alternative Mediterranean Diet Score in 14,815 men and women aged 42–82 years, stratified by sex ^1^.

Sex	Quintile	BUA (dB/MHz)		VOS (m/s)		FFM^BMI^ (kg/(kg/m^2^))	
		*n* =	Mean (95% CI)	*n* =	Mean (95% CI)	*n*=	Mean (95% CI)
**All**	Q1	2725	79.5 (78.9,80.1)	2725	1632 (1631,1634)	2711	1.87 (1.86,1.88)
	Q2	2865	79.3 (78.7,79.8)	2865	1632 (1631,1634)	2848	1.88 (1.87,1.89)
	Q3	3240	80.2 (79.7,80.7)	3240	1634 (1633,1636)	3212	1.90 (1.89,1.91)
	Q4	2928	80.1 (79.5,80.7)	2928	1634 (1633,1635)	2914	1.90 (1.89,1.91)
	Q5	3057	80.6 (80.0,81.1)	3057	1635 (1633,1636)	3035	1.92 (1.91,1.93)
	Q5–Q1		1.04 (0.22,1.87)		2.26 (0.27,4.24)		0.05 (0.04,0.06)
	*p* =		<0.01		0.01		<0.01
**Men**	Q1	1157	89.9 (88.9,90.9)	1157	1644 (1642,1646)	1155	2.30 (2.28,2.31)
	Q2	1213	89.6 (88.6,90.6)	1213	1644 (1642,1646)	1208	2.31 (2.29,2.32)
	Q3	1442	90.0 (89.1,90.9)	1442	1645 (1643,1648)	1423	2.31 (2.30,2.33)
	Q4	1303	90.2 (89.3,91.2)	1303	1645 (1643,1647)	1299	2.33 (2.31,2.34)
	Q5	1375	90.5 (89.5,91.4)	1375	1647 (1645,1649)	1361	2.34 (2.33,2.36)
	Q5–Q1		0.58 (−0.80, 1.97)		2.87 (−0.27, 6.01)		0.05 (0.03, 0.07)
	*p* =		0.25		0.06		<0.01
**Women**	Q1	1568	71.5 (70.8,72.2)	1568	1623 (1622,1625)	1556	1.54 (1.53,1.55)
	Q2	1652	71.4 (70.7,72.0)	1652	1624 (1622,1625)	1640	1.55 (1.54,1.57)
	Q3	1798	72.5 (71.9,73.2)	1798	1626 (1624,1627)	1789	1.57 (1.56,1.59)
	Q4	1625	72.1 (71.4,72.8)	1625	1625 (1623,1627)	1615	1.57 (1.56,1.58)
	Q5	1682	72.7 (72.0,73.4)	1682	1625 (1623,1627)	1674	1.59 (1.58,1.60)
	Q5–Q1		1.19 (0.20, 2.18)		1.44 (−1.07, 3.94)		0.05 (0.03, 0.07)
	*p* =		<0.01		0.14		<0.01

**^1^** Values are adjusted means (95% CI), *n* = 14,815. Means were adjusted for sex, age, BMI, smoking, physical activity, family history of osteoporosis, calcium intakes, supplement use (vitamin D or calcium), medication use (corticosteroids, aspirin or hormone replacement therapy), menopausal status, days of dietary intake data and the ratio of energy intake to estimated energy requirements. BMI was not included in the FFM^BMI^ model. *p* values are for trends calculated using ANCOVA. Range of aMED in each quintile was Q1 = 0 to 2 points, Q2 = 3 points. Q3 = 4 points, Q4 = 5 points, Q5 = 6 to 9 points. BUA = broadband ultrasound attenuation; VOS = velocity of sound; FFM^BMI^ = fat free mass adjusted for BMI.

**Table 4 nutrients-12-01154-t004:** Measures of broadband ultrasound attenuation, velocity of sound and fat free mass by quintile of Mediterranean Diet Score in 14,815 men and women aged 42–82 years, stratified by sex ^1.^

Sex	Quintile	BUA (dB/MHz)		VOS (m/s)		FFM^BMI^ (kg/(kg/m^2^))	
		*n* =	Mean (95% CI)	*n* =	Mean (95% CI)	*n* =	Mean (95% CI)
**All**	Q1	1872	79.1 (78.4,79.8)	1872	1632 (1630,1633)	1852	1.88 (1.87,1.89)
	Q2	2669	79.8 (79.2,80.4)	2669	1634 (1632,1635)	2668	1.89 (1.88,1.90)
	Q3	3468	79.9 (79.3,80.4)	3468	1633 (1632,1635)	3446	1.90 (1.89,1.90)
	Q4	3350	80.1 (79.5,80.6)	3350	1634 (1633,1635)	3314	1.89 (1.88,1.90)
	Q5	3456	80.5 (79.9,81.0)	3456	1634 (1633,1636)	3440	1.91 (1.90,1.92)
	Q5–Q1		1.35 (0.45, 2.25)		2.57 (0.41, 4.73)		0.03 (0.01, 0.04)
	*p* =		<0.01		0.05		<0.01
**Men**	Q1	754	88.9 (87.6,90.1)	754	1643 (1640,1646)	749	2.31 (2.29,2.33)
	Q2	1122	90.1 (89.0,91.1)	1122	1646 (1643,1648)	1123	2.31 (2.29,2.32)
	Q3	1511	90.4 (89.5,91.2)	1511	1646 (1644,1648)	1497	2.32 (2.30,2.33)
	Q4	1504	89.6 (88.8,90.5)	1504	1644 (1642,1646)	1486	2.32 (2.31,2.33)
	Q5	1599	90.7 (89.9,91.6)	1599	1647 (1645,1649)	1591	2.33 (2.32,2.34)
	Q5–Q1		1.85 (0.33, 3.37)		3.95 (0.50, 7.40)		0.02 (0.00, 0.05)
	*p* =		0.08		0.14		0.02
**Women**	Q1	1118	71.6 (70.8,72.4)	1118	1623 (1621,1625)	1103	1.55 (1.53,1.56)
	Q2	1547	71.8 (71.1,72.5)	1547	1625 (1623,1627)	1545	1.56 (1.54,1.57)
	Q3	1957	71.6 (71.0,72.2)	1957	1624 (1622,1625)	1949	1.57 (1.56,1.58)
	Q4	1846	72.6 (72.0,73.3)	1846	1626 (1625,1628)	1828	1.56 (1.55,1.57)
	Q5	1857	72.5 (71.8,73.1)	1857	1624 (1623,1626)	1849	1.58 (1.57,1.59)
	Q5–Q1		0.87 (−0.20, 1.93)		1.34 (−1.35, 4.04)		0.03 (0.01, 0.05)
	*p* =		0.02		0.21		<0.01

^1^ Values are adjusted means (95% CI), *n* = 14,815. Means were adjusted for sex, age, BMI, smoking, physical activity, family history of osteoporosis, calcium intakes, supplement use (vitamin D or calcium), medication use (corticosteroids, aspirin or hormone replacement therapy), menopausal status, days of dietary intake data and the ratio of energy intake to estimated energy requirements. BMI was not included in the FFM^BMI^ model. *p* values are for trends calculated using ANCOVA. Range of MDS in each quintile was Q1 = 0 to 2 points, Q2 = 3 points. Q3 = 4 points, Q4 = 5 points, Q5 = 6 to 9 points. BUA = broadband ultrasound attenuation; VOS = velocity of sound; FFM^BMI^ = fat free mass adjusted for BMI.

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
