# Peer review of "A Mediterranean Diet Is Positively Associated with Bone and Muscle Health in a Non-Mediterranean Region in 25,450 Men and Women from EPIC-Norfolk"

_nutrients, 2020, doi:10.3390/nu12041154_

Round 1

Reviewer 1 Report

The article is interesting and novel, its writing is correct and simple, but it has some aspects to improve:
- Line 72 indicates that the age of the study subjects is between 40-79 years. However, in tables S1, S2 and Figure 2, an age between 39-79 years is indicated. Please clarify.
- A follow-up of 17.4 years is discussed in the results. However, it is not clear from the methodology whether only a single measurement of the diet was made or not. It is precisely a habit that changes greatly over time and age. Neither is it specified how the recording of fractures was done. Please specify better the number of records of information made.
- Although it is not the aim of the study, it would be advisable to carry out analyses stratified by age groups. And in women by menopausal status.
- I recommend including Figure S1 in the paper, because it better clarifies the methodology and the sample used.

Author Response

Reviewer 1:

  1. Line 72 indicates that the age of the study subjects is between 40-79 years. However, in tables S1, S2 and Figure 2, an age between 39-79 years is indicated. Please clarify.

We have now corrected line 72 to 39-79 years.

  1. A follow-up of 17.4 years is discussed in the results. However, it is not clear from the methodology whether only a single measurement of the diet was made or not. It is precisely a habit that changes greatly over time and age. Neither is it specified how the recording of fractures was done. Please specify better the number of records of information made.

We have now clarified that diet was a single measurement: “Food and nutrient intakes were estimated from seven-day food diaries completed at one time point (first health check (1993-1997)” (Line 101).  We have also provided further details of how fractures were ascertained: “Osteoporotic fractures were identified using the East Norfolk Health Authority database which records all hospital contact that Norfolk residents have in England and Wales.  International Classification of Disease codes 9 and 10 diagnostic codes were used to ascertain fractures by site.  Data were available to 31st March 2016.  We defined total fractures as any incident fracture at the hip, spine or wrist” (Line 95).  

  1. Although it is not the aim of the study, it would be advisable to carry out analyses stratified by age groups. And in women by menopausal status.

We repeated the results of our main fracture analysis in participants aged over 50 years, these results are presented in Figure 2.  As the fracture rate in participants aged <50 years was so low (<8% of total fractures) we did not conduct analysis in this age group.  We have now conducted the analysis for post-menopausal women and the results are presented in Figure S1.

  1. I recommend including Figure S1 in the paper, because it better clarifies the methodology and the sample used.

We thank the reviewer for this suggestion.  We have now included figure S1 in the main paper.

Reviewer 2 Report

This study examined the relationship between adherence to the Mediterranean diet and fracture risk in a UK population. The study benefits from large sample size and a longitudinal follow up, which provide convincing results to support the hypothesis. I have some comments on the methodology and the write up of the manuscript:  1. Title: The word 'musculoskeletal' is very board, covering bone, muscle and joint. The current manuscript covers only the bone and muscle aspects, but not the joint. So the title should be amended to reflect the content.  2. Introduction: Paragraph 1 is too short to stand alone and should be combined with paragraph 2.  3. The authors should highlight the research question and gap in the current literature in the Introduction.  4. Materials and Methods: Please provide the ethical approval code.  5. The details of quantitative ultrasound (QUS) measurement of bone are too brief. Which type of QUS is CUBA (gel or water-based)? What is the short-term in vivo coefficient of variance of the machine? Which foot is used for measurement? How did the authors normalise the variation in QUS measurement? What is the agreement of the machine with DXA results? Does the machine generate a composite index using VOS and BUA values, like estimated BMD or stiffness index? 6. Similarly, details of the body composition measurement are vague. How many electrodes does the machine have? Its coefficient of variance? Agreement with DXA body composition estimates? 7. What is the disease code used for fracture? 8. Table 1: superscript for kg/m2. What is the meaning of 'y' in the table? 9. For Table 3 and 4, it will be more informative if the authors could insert the range of each Q  10. For results depicted in Figure 1, were the results adjusted to BUA/VOS or FFM in each model? 11. For all figures, please put the legend after the figures.  12. Line 332, stratification based on 'sex', not gender 13. Discussion: Can the authors elaborate how do the components in the Mediterranean diet exert individual and synergistic effects in preserving bone and muscle mass, and subsequently prevent fracture? 13. Discussion: Limitation of the study: Please provide evidence that the QUS and BIA devices used in the study are equivalent with DXA, rather than citing results from different devices. 

Author Response

Reviewer 2:

  1. Title: The word 'musculoskeletal' is very board, covering bone, muscle and joint. The current manuscript covers only the bone and muscle aspects, but not the joint. So the title should be amended to reflect the content.

We have now changed the title to: “A Mediterranean diet is positively associated with bone and muscle health in a non-Mediterranean region in 25,450 men and women from EPIC-Norfolk”

  1. Introduction: Paragraph 1 is too short to stand alone and should be combined with paragraph 2.

Correction made (Line 39).

  1. The authors should highlight the research question and gap in the current literature in the Introduction.

We have added the following to the introduction: “The aim of the current study was therefore to examine associations between adherence to a Mediterranean diet and key indicators of bone and muscle status, including fracture incidence, bone density and fat free mass in a large longitudinal cohort of 25,450 men and women living in the UK.  We also examined the associations between our bone and muscle status indicators and fracture risk in order to ascertain their value for prediction of fracture risk with a longer follow-up period than previous reported.  Earlier studies examining the associations between Mediterranean diet adherence, bone density and FFM have been restricted by limited sample sizes [15-17].  In addition this will be the first study to use food diaries to examine Mediterranean diet adherence in a large longitudinal study of fracture risk” (Line 58).    

  1. Materials and Methods: Please provide the ethical approval code.

The ethical approval code has been added (Line 74)

  1. The details of quantitative ultrasound (QUS) measurement of bone are too brief. Which type of QUS is CUBA (gel or water-based)? What is the short-term in vivo coefficient of variance of the machine? Which foot is used for measurement? How did the authors normalise the variation in QUS measurement? What is the agreement of the machine with DXA results? Does the machine generate a composite index using VOS and BUA values, like estimated BMD or stiffness index?

We have now added further details of the QUS measurements, unfortunately a composite index was not available: “Measurements of broadband ultrasound attenuation (BUA in dB/MHz) and velocity of sound (VOS in m/s) were taken in duplicate from each foot. The mean of the left and right measurements were used for analysis.  The coefficient of variation was 3.5%.  Five CUBA machines were used and calibrated daily with a physical phantom.  The machines were also compared on one calcaneus and whilst significant variations between machines arose adjustment for temperature and machine were not found to change BUA-fracture relations [16].  Analysis in this cohort also showed that the power of QUS for prediction of fractures is comparable to that of DXA with the hazard ratio for a 1 SD decrease in DXA-measured total hip BMD was 2.3 (95% CI: 1.7–3.0) compared with 2.0 (95% CI: 1.6–2.7) for a 1 SD decrease in BUA [17]” (Line 114).

  1. Similarly, details of the body composition measurement are vague. How many electrodes does the machine have? Its coefficient of variance? Agreement with DXA body composition estimates?

We have added in the following further information about the BIA measurements: “Bioelectrical impedance analysis (BIA) was also undertaken using a standardised technique (Tanita TBF-531, Bodystat, Isle of Man, UK).  The machine had two toe and two heel electrodes” (Line 124) and “The correlation between DXA and BIA calculated fat free mass are high (0.95) [19]” (Line 133).  

  1. What is the disease code used for fracture?

Further details of the disease codes have now been added: “International Classification of Disease codes 9 and 10 diagnostic codes were used to ascertain fractures by site” (Line 127)

  1. Table 1: superscript for kg/m2. What is the meaning of 'y' in the table?

Corrections made (Table 1).

  1. For Table 3 and 4, it will be more informative if the authors could insert the range of each Q

The ranges of each score now been added (Lines 384 and 407).

  1. For results depicted in Figure 1, were the results adjusted to BUA/VOS or FFM in each model?

The results in Figure 1 (now Figure 2) were to show the relative association of each parameter with fracture risk so we did not include all outcomes in the same model.

  1. For all figures, please put the legend after the figures.

Correction made

  1. Line 332, stratification based on 'sex', not gender

Correction made.

  1. Discussion: Can the authors elaborate how do the components in the Mediterranean diet exert individual and synergistic effects in preserving bone and muscle mass, and subsequently prevent fracture?

We have added the following which we hopes clarifies our meaning: “Whilst these dietary constituents are important components of the Mediterranean diet and have mechanistic links with musculoskeletal health, this research suggests that the synergistic effects of these, and other individual components, are likely to be of greater importance to reducing fracture risk.  Previous research has shown that the food matrix exhibits distinct relations with health outcomes compared to single nutrients, for example, whole dairy products have shown to have a greater effect on bone mass than single dairy constituents, such as calcium and vitamin D [31]”.

  1. Discussion: Limitation of the study: Please provide evidence that the QUS and BIA devices used in the study are equivalent with DXA, rather than citing results from different devices.

We have added the following to show the QUA device is equivalent to DXA in this cohort: “and produces comparable fracture risk predictions as DXA in this cohort [17]” (Line 493).  We have also added further information regarding the comparison between QUS and DXA in this cohort to the methods (please see comment 5).

Round 2

Reviewer 2 Report

I have no further comments for this manuscript. My previous comments have been addressed. Thank you.